# Experiences of Middle School Programming in an Online Learning Environment

**DOI:** 10.3390/bs12110466

**Published:** 2022-11-21

**Authors:** Reem Alebaikan, Hayat Alajlan, Ahmad Almassaad, Norah Alshamri, Yvonne Bain

**Affiliations:** 1Department of Curriculum and Instruction, College of Education, King Saud University, Riyadh 11451, Saudi Arabia; 2Ministry of Education, Riyadh 11148, Saudi Arabia; 3School of Education, University of Aberdeen, Aberdeen AB24 3FX, Scotland, UK

**Keywords:** inclusive education, social and emotional learning, computer coding, informal learning, K-12 education, online learning

## Abstract

This small-scale qualitative study aimed to explore learning programming through online experiences among middle school students in a school for girls in Saudi Arabia. The low uptake of computing by girls has been a persistent problem in schools and beyond. In Saudi Arabia, there are similar issues in encouraging learners and girls in particular to be interested in computer coding. To explore how to engage learners in coding, an informal online course on programming for learners (age 12) was designed using a community-of-inquiry approach and a gamification process enabled through the use of Thunkable™ and TalentLMS™ to engage learners. An inductive qualitative research approach was used to explore influencing factors for engaging learners in programming. The data comprised three individual interviews, one focus group, a teacher’s diary, and a content analysis of the activities recorded in the TalentLMS™ system’s student progress reports. Findings highlighted the need to consider digital learning agency in the online learning environment and that learning programming online was best facilitated through student collaboration using live tools with teacher support to develop the online community. Further, findings revealed the rationale for the girls enrolling in the online programming course, which included improving their online learning skills, planning future careers, and developing résumés. These findings may contribute to offering some insight into pedagogy that can encourage greater interest in computer programming in schools.

## 1. Introduction

This small-scale qualitative study aimed to explore learning programming through online experiences among middle school students in a school for girls in Saudi Arabia. The low uptake of computing by girls has been a persistent problem in schools and beyond. In Saudi Arabia, there are similar issues in encouraging learners and girls in particular to be interested in computer coding. To explore how to engage learners in coding, an informal online course on programming for learners (age 12) was designed using a community-of-inquiry approach and a gamification process to engage learners.

### 1.1. Context for the Study

The context for the study was a private girls’ middle school in Saudi Arabia. In Saudi Arabia, schooling is gender-segregated and girls are taught by female teachers. Schools typically serve a particular age range from 6 to 17 years, with middle schools for age 12 to 14 years and high schools for age 15 to 17 years. Schools may be state-organized or private. At the time of the study, the middle school curriculum in Saudi Arabia did not include programming, although schools could incorporate some basic elements of coding into their curriculum. However, very recent curriculum changes now mean that programming is mandatory. Learners in both male and female and middle and high schools will have compulsory computing courses that include programming.

At the time of the study, as coding had not been introduced to the curriculum, an online informal learning course was created to explore how middle school learners might be encouraged to engage in programming and develop an interest in coding. The online experience was designed to enable flexible provision of the informal course. The course was purposively created to research a possible community of inquiry (COI) and games-based pedagogy that may engage learners in an online course about coding and to provide insight of the girls’ rationale for engaging with this.

### 1.2. Online Learning in Schools

Online learning, in which students learn through teacher and peer interactions in synchronous or asynchronous environments from anywhere and at any time using devices connected through the internet [1], has become an increasing trend in K-12 education [2,3,4]. Whilst the COVID-19 pandemic led to a rapid shift to online learning for school learners, online learning had already been established in schools, for example to supplement or replace face-to-face courses for students who may have failed courses or to access courses not offered at the students’ current school [2]. However, the effectiveness of online learning for various K-12 courses requires evaluation, especially for the more practical subjects that would normally involve lab work, collaboration, and teacher guidance based on face-to-face activities, such as studying programming languages. This research aims to provide an evaluation of the learners’ experiences of studying programming online.

While online environments may offer autonomy and empowerment to students [5], the learning needs to be meaningful and substantive to fully engage the students in online learning. Garrison, Anderson and Archer [6] have long argued for creating meaningful learning through a collaborative COI approach for which cognition and teaching presence were found to be important influences on learning and satisfaction, whilst social presence was associated with student satisfaction [7]. Grover, Pea, and Cooper [8] similarly draw attention to online engagement, noting that both the cognitive and affective dimensions need to be considered as student abilities to transfer learning to online contexts (cognitive dimension) and are associated with the depth of the learning concepts and constructs, and the student attitudes toward using online tools (affective dimension) are related to their curriculum interest and awareness and their own self-discipline.

Additional consideration of influences on student completion of online courses highlights the need for more nuanced research [4]. In relation to learning programming, online K-12 programming course dropout rates have been found to be very high, with student persistence being the highest in those that had prior programming knowledge and/or an intrinsic interest in the subject matter [9]. Further, as online programming course delivery is still a relatively undeveloped area [10], little is known about the student perceptions regarding the support being provided by the online teachers [2]. Other online learning student perception issues have been found to be related to understanding the motivating factors that influence a student’s decision to engage in the class [11].

## 2. Designing the Learning Experiences

There has been a recent increase in research on the design, delivery, and support for K-12 online learning [12,13]. Borup, Chambers and Stimson [2] claimed that there were three common challenges to online learning: a high degree of flexibility, technological competence, and online communication. Having a high degree of flexibility was seen to be an obstacle to course completion, and technological competence was related to the quantity of instructions needed for students to access online courses. Hendrix and Degner [14] reported that high school students found it difficult to communicate with online teachers. A recent study [15] identified that communication in an online learning environment required that three basic conditions be met: continuous encouragement, timely feedback, and a sense of enthusiastic identification through competition and performance assessment. In the design of the course created, this was enabled through scheduling regular points of interaction with the teacher and facilitating peer-group learning activities to solve coding puzzles.

Subject or activity integration has also been found to influence the educational impact of computer programming [16]. Further, there is recognition that more initiatives are required to support teachers in developing appropriate pedagogy for learning programming [17]. Programming learning requires clear instructional activities [18]. Popat and Starkey [19] (p. 365) found that learning coding had a number of positive educational outcomes, such as “mathematical problem-solving, critical thinking, social skills, self-management, and academic skills”.

Settle, Vihavainen and Miller [10] reviewed research on online programming education and suggested that courses should: utilize appropriate web features including interactive tools and media; encourage interaction and collaboration (using LMS or other tools) to facilitate student engagement; provide timely feedback; and ensure that task design is informative. In online education, student engagement through active learning may be achieved by employing a pedagogy with learning strategies such as storytelling, gaming, simulation, and pair-programming approaches [20]. Game-based learning has often been used to facilitate student engagement [21,22]. Therefore, learning strategies such as gaming, simulation, and pair-programming approaches could assist in developing student creative thinking, which could assist them in being future producers rather than consumers of technology [20].

Further support for this approach comes from Pilkington [23], who suggests that gamified environments may have positive motivational impact on programming learning. Other studies have suggested that gamification elements be integrated into online learning platforms [24] as these could enhance the students’ collaboration, communication, and problem-solving skills [25]. While the benefits of gamification in education have been recognized, there is no agreed gamification process design [26]. However, Piteira, Costa and Aparicio [26] proposed a gamification framework to guide teachers of online distance programming courses that was based on the following pillars: course general goals, learning outcomes, topics, contents, gamification, cognitive absorption and flow, and personality.

When teaching programming, the tutor needs to have constructivist attributes that facilitate the learner’s ability to solve problems and reflect on their performances [21]. Specifically, teachers in middle and high school need to motivate students to engage in the classwork, maintain their attention using various student-centered strategies, and understand what motivates each individual student; however, these are not always successful [11]. Other studies [27,28] highlight that gamification with a community-of-inquiry approach brings benefits to student learning.

This study incorporated the community of inquiry (COI) framework as a theoretical framework to enable greater research analysis breadth [29]. COI is a collaborative–constructivist approach to a learning framework (Figure 1) from which to study the dynamics of online learning communities [9,30]. The COI framework focuses on three elements and the dynamics of these elements required for an online educational experience: social presence, teaching presence, and cognitive presence [31].

These three elements facilitate the creation of collaborative communities of learners “engaged in exploring, creating meaning, and confirming understanding” [30] (p. 352). The COI framework, therefore, provides a dynamic model for transforming what could be a passive online experience into an engaging educational experience based on collaborative thinking and learning [32]. As one of the online learning challenges is the technology versus the learning, COI enables the focus to be on the learning rather than on the technology [30].

### The Online Learning Environment

As the research aim was to explore the online experience of learning programming, a block-based coding platform, Thunkable, was selected (Figure 2). Using drag and drop, Thunkable allows the students to build their own game through setting layout, adding various components such as buttons and images, adding voices, and downloading, sharing and publishing the game. Thunkable was recommended by Common Sense Education for grades 6–12 and described as having the potential to integrate design and computational thinking [33]. Common Sense Education is a nonprofit organization that supports pre-K-12 schools with lesson plans and EdTech reviews to help students thrive in a connected world.

Gamification principles, employed through the use of TalentLMS, were therefore embedded in the course design to foster student engagement and encourage interaction [26]. Several learning management systems were first reviewed to determine the best features to promote gamification, with TalentLMS being finally selected because of its points, badges, levels, and rewards gamification techniques. Therefore, TalentLMS was used to provide the digital materials and facilitate the communication between the students and the teacher. Arabic was used for the content and guidelines as well as being the first language of learners for interaction. Five stages of gamification were developed in TalentLMS, with the first being the orientation stage (Figure 3). A stages map was available for the students to follow their progress.

The course required the students to access the Thunkable and TalentLMS platforms separately. In Thunkable, students practice programming to develop a simple “Super Mario” game, guided by the stages map on the TalentLMS platform. In TalentLMS, students add a URL link of their performance of Super Mario in each stage. Students are allowed to transfer from stage to stage by teacher approval, and their achievements are rewarded by points and badges automatically.

Significantly, the course design was underpinned by the K-12 National Standards for Quality Online Courses [34], which were built based on the work by the iNACOL National Standards for Quality Online Courses, one of the most popular sets of standards [35]. All national standards for quality online course areas (course overview and support, content, instructional design, learner assessment, accessibility and usability, technology, and course evaluation) were met. The instructional course design was aimed at fostering interaction and promoting a deep engagement between the students, with the formative and summative assessments being included as activities or quizzes. Synchronous and asynchronous interactive online learning environment features were employed to provide programming activities, online quizzes, online discussions, and live sessions. The virtual classrooms allowed for synchronous interaction via a chat area and/or by using a microphone. The virtual classroom is included in TalentLMS (see Table 1). The live sessions and online discussions would create the opportunity for peer-to-peer discussion as well as teacher–learner interaction, thereby enabling the potential development of the social presence and teacher presence. The cognitive presence was potentially developed within the gamification approach of Thunkable as well as through the discussions and live interactions.

The course design and implementation were developed by three computer teachers, including the teacher who participated in the study. The course design, duration and teacher role were discussed in detail with the course teacher before the experiment to ensure that the LMS and programming platform selections were suitable for the students’ ages and were easy to use. The final course design and the materials were reviewed by three experts and then modified based on their suggestions, after which a pilot study was conducted with three individual students (different from the main study sample) to ensure course design clarity and determine whether the materials needed further modification. While the research was being conducted, the authors conducted ongoing evaluations of the experiences.

The course design process took four months and was designed as an informal extracurricular online learning experience to learn programming, created for the purpose of the study. The learning experiences included one to two weekly sessions of 2 to 3 h of online synchronous activity (six to nine synchronous activities) and 3 to 4 h (nine to twelve activities) of asynchronous activity over a period of 3 weeks.

The course announcement, which was delivered via a video clip attached to participant information and parent consent forms, described the course aims, the requirements, the duration, and the teacher’s contact information and also described the motivational rewards to encourage course completion. The video, which had an animation made within the Thunkable environment, invited the students to participate in the programming course by developing a game on the Thunkable platform, discussed the required time needed per week, encouraged a flexible approach, and gave information on the teacher support available in the live sessions. After receiving the students’ information and the parents’ consent forms, the students were sent acceptance emails with the course guidelines.

## 3. Research Methodology

The purpose of the research was to explore the middle school students’ online programming course experiences in this purposefully created informal setting, driven by the following research questions:What is the rationale for enrollment in online programming courses?What are the main influencing factors to learn programming effectively as an online experience?

### 3.1. Research Design

This study used an interpretative paradigm to understand and interpret the practices and experiences of middle school students in online programming. The research questions and the methodology of this study led to use qualitative approaches that are more effective in exploring subjective meanings within a culture, understanding perceptions of participants [36]. Therefore, qualitative methods were used to obtain rich descriptive data to facilitate the exploration of the phenomena.

### 3.2. Participants

A purposive sampling method was used to select the students and an instructor for this study. All participants and the instructor were female due to the segregated nature of the Saudi context.

The main criterion for the instructor was teaching and academic experience in computer education. Therefore, the selected participant teacher was a computer teacher with a master’s degree in computer curriculum and instruction with 15 years of teaching experience in public schools. Moreover, the instructor had assisted in the design of the online learning experience.

The main selection criteria for the participants students were that they were at the same middle school, had internet access, were familiar with programming concepts and had some experience of learning programming (formally or informally), such as accessing the “hour of code” initiatives [37] and that it would be possible to access parents. The school selected was not that in which the instructor taught. The sampling process for the learners started by reaching out to a group of mothers of 16 female students through the WhatsApp application to explain the purpose of the study and send the course announcement. Thirteen students agreed to join the research and participated in this study.

### 3.3. Data Collection Methods

A pragmatic approach was used to decide which data would be gathered and how the research questions could be addressed. Data were gathered in 2019, pre-COVID pandemic. One of the ways to bring credibility to a qualitative study is through triangulation [37,38]. Therefore, four methods were used to collect data: individual interviews, focus group, diary, and content analysis. The data were collected from one individual interview with the teacher, three individual interviews with the students, one focus group (six students), the teacher’s diary, and a content analysis of the activities and the system’s student progress reports.

### 3.4. Ethical Consideration

Ethical research approval was received from the Ethics of Human and Social Research Committee of King Saud University (KSU) to conduct this study (KSU-HE-21-371, date of approval 10 June 2021). Consent was given by the teacher, the students, and the students’ parents, and the students and the teacher were given netiquette documents. All participants were told that anonymity in the research was guaranteed and that they had the right to withdraw at any time without giving a reason.

### 3.5. Data Analysis

All interviews and the focus group discussion were recorded and analyzed anonymously. The teacher sent her diary to the authors through email. The students’ progress reports were printed from TalentLMS. As the data were in Arabic, they were first coded then translated into English, with the transcription and coding each being undertaken by two researchers to ensure results agreement. The key themes were identified after the initial coding and discussed by the research team, and quotations were selected from the data to provide readers with sufficient knowledge and enable transferability judgment [38].

## 4. Findings and Discussion

### 4.1. Rationale for the Online Programming Course

The most significant findings for the first research question are detailed in the following. It was found that the participants had positive attitudes toward the online programming course and realized the benefits. The participants highlighted several advantages of enrolling in the online programming course, including: developing their problem-solving skills; investing their time; being responsible; a passion to learn programming; planning for their future careers; and developing their resumes. One participant said, “I will encourage my friend to attend such a programming course, I will tell her she will get benefits and have a new experience and learn about programming”. The passion to learn new things and be able to develop games was mentioned by one of the students, and she added, “Probably I will pursue game design in the future”.

One student said that the online programming course had helped develop her problem-solving skills, which was in line with Popat and Starkey’s [19] findings that learning coding positively reflected on student problem-solving skills and appeared to confirm that both classroom and online learning environments had the ability to promote problem-solving skills. The students also emphasized that the gaining of the certificates and awards could help their future careers.

Participants emphasized that the online course helped them develop online learning skills, which was in line with Lowes and Lin’s [39] finding that learning a subject online also strengthened online learning skills. The students appreciated the flexibility, the ubiquity of the online learning, and the collaborative opportunities, with most describing the experience as enjoyable and interesting and having a high social presence level because they were able “to identify with the community, communicate purposefully in a trusting environment and develop inter-personal relationships by way of projecting their individual personalities” [30] (p. 352).

### 4.2. Factors Influencing the Effectiveness of Online Programming Learning

The most significant findings for the second research question are detailed in the following. From the analysis of the rich data gathered from the participating teacher and students (interviews, focus group, teacher’s diary), the learning-to-program online experience was divided into four main themes, which were then formulated into the online programming learning environment framework for middle school students (Figure 4). One important factor in this framework was the rationale for the online learning discussed in the previous section and the three other factors were the course design, the teacher’s role, and the digital learning agency, which are described in the following sections.

### 4.3. Course Design

In general, the students were satisfied with the course and the digital materials (texts, flowcharts, and videos), support and guidelines and the live sessions (virtual classrooms). One student commented that “The course increased my enthusiasm to learn programming because the only other way is to learn programming is through YouTube, which does not help me practice. This course offers guidelines and support from the teacher as well as enabling me to practice”. This finding highlighted the importance of the need for communication and support, that is, the cognitive presence, which “is the extent to which learners are able to construct meaning through sustained communication” [40] (p. 24).

Significantly, the course design was underpinned by gamification principles, the value of which was acknowledged by the participants, with the competitive learning context being appreciated. The students liked the competition gamification principle, with one commenting that “online learning has great advantages. It is enjoyable and interesting when it includes competitions. It is fabulous and great”. In addition, the teacher claimed, in the interview, that the gamification increased the students’ motivation and enthusiasm to get more points and badges and reach the last stage to earn the trophy. This result emphasized the positive impact of gamified learning in increasing learner engagement and motivation [41,42,43].

Moreover, the construction of the Super Mario game using Thunkable motivated the students, as this was mentioned in all individual interviews as well as in the focus group. One student considered that learning programming in the online environment was difficult at the beginning. However, she clearly stated that her programming skills developed: “In the last stages, I mastered it and it was not difficult”. She continued: “I created a game and played with it”. Another student commented that “the course enabled me to learn enjoyable things. I have learned how to develop a game. I learn the fundamental issues in programming with fun as I made a game and played with it”. This finding aligns with previous findings that learning basic programming concepts via game development can be enjoyable and can increase positive attitudes toward the work [44].

Feedback from students was used to modify their ongoing experience, such as adjusting times to account for school exams and incorporating SMS and emails to remind them of activities and live sessions. The students preferred to get support via the live sessions rather than by participating in the online discussions. One student said that “the live sessions helped me understand the difficulties that I met in accessing the link to Thunkable”. The students were very active in the live sessions, with all of them interacting using the chat tools, except one who participated verbally. In the focus group, the students highlighted helpful features of the live sessions. One student commented, “We hear each other, talk or text, this makes many things easier as we help each other”. Another student added, “Similar to PUBG game, we can use texting to guide each other. This way, we can communicate at the same time”.

Very few students used the microphone during the live sessions, and one student clarified the reason behind that was to maintain her family privacy, commenting that “sometimes I can talk in the live session and other times I cannot because my family are around me”. However, not all students participated in the asynchronous online discussions, even though the course instructions encouraged the students to participate in the lesson stages. Therefore, there appeared to be a reluctance to use the discussion forum, which may have been because the students preferred to have support using tools that provided prompt responses. However, the teacher said that as most used text chatting in the live sessions, the online discussion had the potential to be effective. This then raises the question as to whether texting was related to the asynchronous interaction or for other reasons. Further research investigating cultural and online learning tools such as online discussions is needed for clarification.

### 4.4. Teacher’s Role

There is an increasing body of evidence that teaching presence is a key component of successful online learning [45]. In this study, the teacher indicated that adding more live sessions was beneficial and highlighted the importance of teaching presence: “the presence of the teacher [in the live sessions] provides live guiding which enables students to complete their programming tasks simultaneously”. She also stated that “it adds more enthusiasm as the student learns with peers”.

Two students discussed the teaching presence on the programming platform and suggested that there be a virtual assistant, with one commenting that “If I have a virtual assistant guiding me on the programming steps, this would make the tasks easier. I would not need to go back to the explanation in the video to find the missed steps”. She mentioned the availability of this feature in games and reflected on its usefulness in programming learning.

The findings of this study revealed elements for effective online teaching from the student perspective. In particular, the participants emphasized the need for synchronous teacher support in virtual meetings to answer questions and provide orientation, which highlighted the importance of student–teacher interaction in ensuring online K-12 education completion rates [45] and the essential aspect of the teaching presence for a community of inquiry.

The teacher emphasized that the online programming gamification design influenced the student engagement and felt that the online programming course had been successful. However, she suggested that a face-to-face orientation or a mandatory live orientation session would assist in fully clarifying the course objectives and course plan, saying that “I think having a live orientation meeting in the beginning of the course would address the main challenges facing the students”. The teacher was also aware of the students’ need for timely support and replies to their inquiries, saying that the students expected the teacher to respond promptly with a clear, concise message and wanted to be informed about how long it would take to receive replies to their inquiries. These findings emphasized the three online learning environment communication requirements: encouragement, timely student feedback, and enthusiastic identification via competition and performance [15].

The need to maintain the COI “teaching presence” element was further identified in the results. For example, the teacher noticed that students were more motivated when they received collaborative communicative encouragement during the live sessions and system messages. This finding contradicted previous findings that there was less student-teacher interaction in online learning environments than in a traditional face-to-face environment [46]. Fessakis, Gouli and Mavroudi [47] concluded that when learning programming, teachers need to reinforce interactivity in the collaborative activities.

### 4.5. Digital Learning Agency

A key finding that emerged highlighted that digital learning agency needs to be enabled for middle school online learning. The students demonstrated high-level abilities to deal with the new technologies, affirming Passey et al.’s [48] digital agency belief that people are able to adopt, adapt and use technologies wisely and responsively. Agency is defined as “the capability of individual human beings to make choices and act on these choices in a way that makes a difference in their lives” [49] (p. 135). Digital agency, therefore, is a set of practices related to and arising from learner agency concepts [48], i.e., making learning choices about time, place, learning style (individual or collaborative, learning from watching videos or reading texts), and the technology to be used (laptop or mobile device). For example, one student pointed to the power of choice while learning and suggested that various other types of games be developed in addition to the main Super Mario game.

As described by the teacher, the online learning environment was found to facilitate student-led instruction; therefore, learner agency was enabled through the student-led instruction, which was when the teacher provided the opportunity for the students to prove their agency by leading the instruction on a particular skill or concept [50]. The students used different strategies to learn and accomplish the online programming tasks. For example, students learned by themselves (self-learning skills), supported each other (collaborative online learning) during the live sessions, and offered peer support without needing the teacher’s encouragement. Fessakis, Gouli and Mavroudi [47] (p. 88) concluded that “computer programming environments support autonomous or guided open-ended explorations in the process of which the children participate actively, think and control the computer”. Although the participants demonstrated digital learning agency, the high level of flexibility, which has been recognized as a possible completion obstacle [2], was a challenge for some students, with two of the active students not completing the course requirements. Studies have found that maintaining persistence in online learning environments can be a significant challenge [51]. However, technological competence and online communication were not found to be issues in this study, although it does suggest the need to rethink how teaching and learning online is supported as suggest by Amos et al. [52].

The live collaboration online course tools were found to facilitate student engagement and motivation. The participants stated that they preferred the live collaborative learning over the self-learning, which reinforced the notion that learning agency is when students have “the understanding, ability, and opportunity to be part of learning design and taking action to intervene in the learning process and become effective lifelong learners” [53] (p. 18). During the focus group, the students claimed that accessing the Thunkable platform together in the live sessions enabled them to effectively communicate and collaborate, with one stating that “this (experience) simplified many steps as we helped each other when working at the same time”. The effectiveness of the collaboration in the live sessions was a key finding in this study.

Significantly, it was found that learning in the online environment enabled the three digital agency components [48]. For example, the students’ digital learning practices revealed: their confidence in using the internet (digital confidence); their effective navigation of the digital world (digital competence); and their ability to be responsible for their digital practices (digital accountability). The students connected to the internet with their smart phones or laptops, which demonstrated digital confidence. There were no reports of any login difficulties or problems with their TalentLMS accounts or the Thunkable platform, which reflected their digital competence. Because some expressed concern about their future careers and wished to maintain their family privacy, most opted to use the texting live tools rather than the microphone, which reflected their digital accountability. Table 2 shows examples of quotes/data reforming the subthemes and the main themes.

Garrison [45] considers that the cognitive presence should allow for “exploration, integration and resolution” and often the integration and resolution steps are missing. Further, he considers that the teaching presence has different roles in design facilitation and direction of the online interactions and discussions. The findings in this study suggest that digital learning agency needs to be incorporated into the design and facilitation of online learning experiences.

Research has found that the COI framework can be adjusted to the context. For example, Armellini and Stefani’s [54] study introduced an adjusted COI framework in which the social presence was more prominent in the teaching and cognitive elements. Significantly, the current study reveals that the digital learning agency links the elements of the COI framework, that is, by transferring the passive lectures into educational experiences [32]. This study showed that the educational experiences of the middle school students in an online environment were developed through their digital learning agency and therefore suggests that the educational experiences should explicitly encompass digital learning agency (Figure 5).

The findings in this study offer some exemplification and guidance on how to offer online, on-demand learning to engage girls in programming, supporting the literature that more girls showed interest in learning to program through block-based coding [55]. Whilst the educational challenges presented by COVID-19 encouraged educators to design appropriate responses within specific contexts [56], this pre-COVID study exemplifies online course design choices to engage in learning programming. As most middle school students are digital natives who already have digital learning agency, they are able to be active in shaping their own learning. However, there remain questions over effective asynchronous engagement.

## 5. Conclusions

This research has several contributions. As far as can be determined, this is the first research focused on the teaching of programming online to middle school students that emphasizes digital learning agency within a COI framework. By applying a COI approach to the design and analysis of the online learning experiences, it was found that digital agency needs to be considered as part of the online learning environment for middle school students because, as stated by Grover et al. [8] (p. 224): “the middle school years are crucial for identity building”. The learners’ interactions have revealed a preference for synchronous online engagement, which raises implications for the design and facilitation of online learning for middle school learners. Learning programming online requires students to collaborate using live tools. The commonality of online gaming was found to have influenced middle school student perceptions of the best online interaction environment for learning. As literature shows that gamification is broadly introduced into online learning [57], this research emphasizes the positive effect of gamification on students’ motivation and engagement.

This research has limitations similar to any small-scale qualitative study and the findings may not be broadly generalized because the study was conducted over a short time frame and was dependent on the students’ study schedules. All participants were female students from the same school and class (due to the segregated nature of the context), and there was the possibility of self-selection bias, as the students who joined the course possibly had a relatively high level of interest already in learning programming. However, future directions could seek to overcome these limitations by studying a broader sample of middle school students (including male students) and teachers in different contexts. This recommendation supports the literature on the need for further research to explore the utility and efficacy of block-based coding environments for novice programmers [58]. Future studies could also explore the key limiting factors to effective online learning of programing and possible changes at the educational system level. Significantly, the findings from this empirical study provide some crucial pointers for the development of online middle school programming pedagogy, with perhaps the most salient implications being the need to ensure synchronous peer-collaboration facilitation for programming practice and to consider the further development of digital agency to encompass asynchronous and synchronous engagement and interaction.

## Figures and Tables

**Figure 1 behavsci-12-00466-f001:**
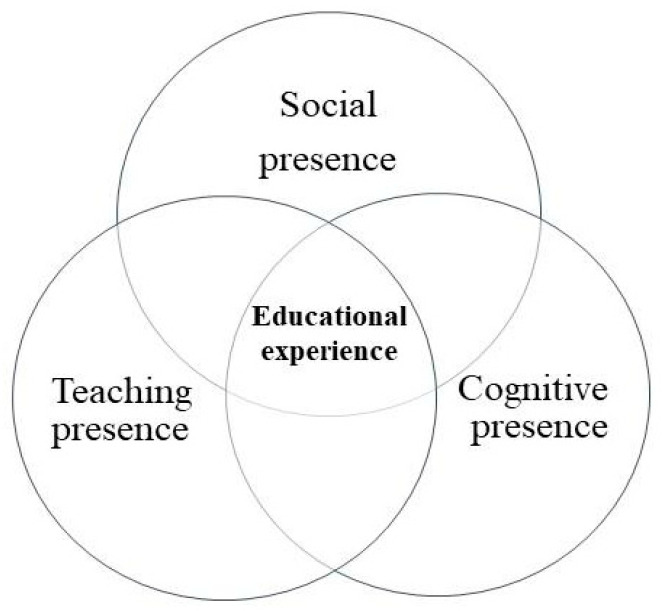
Community-of-inquiry framework [6] (p. 6).

**Figure 2 behavsci-12-00466-f002:**
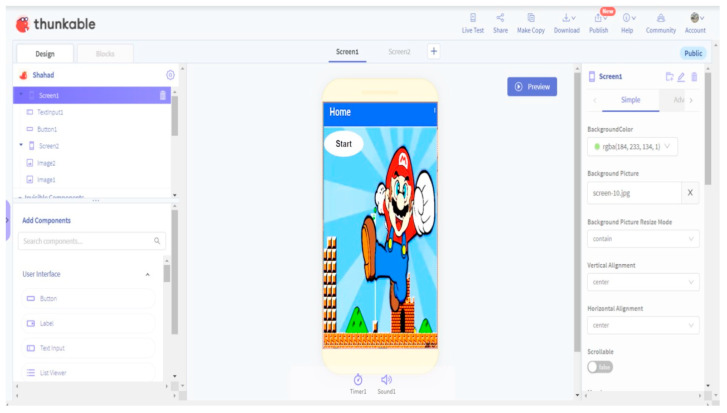
Thunkable user interface.

**Figure 3 behavsci-12-00466-f003:**
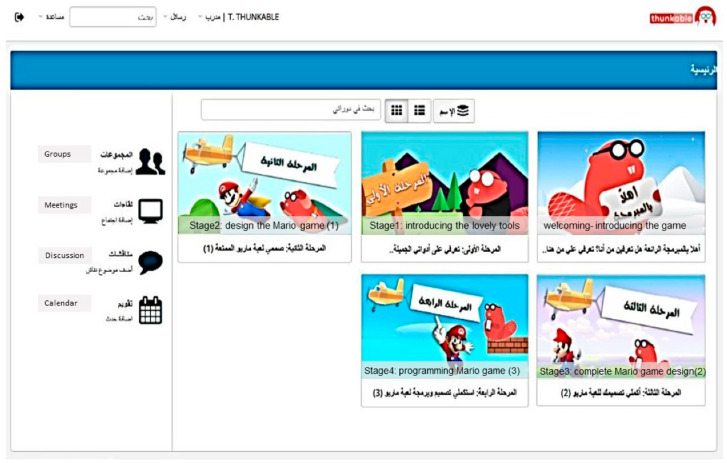
Main TalentLMS interface showing the orientation stage and subsequent four learning stages.

**Figure 4 behavsci-12-00466-f004:**
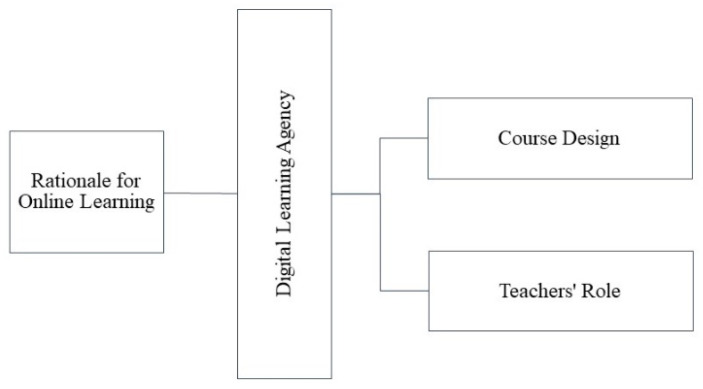
Factors influencing the effectiveness of the online programming learning environment for middle school students.

**Figure 5 behavsci-12-00466-f005:**
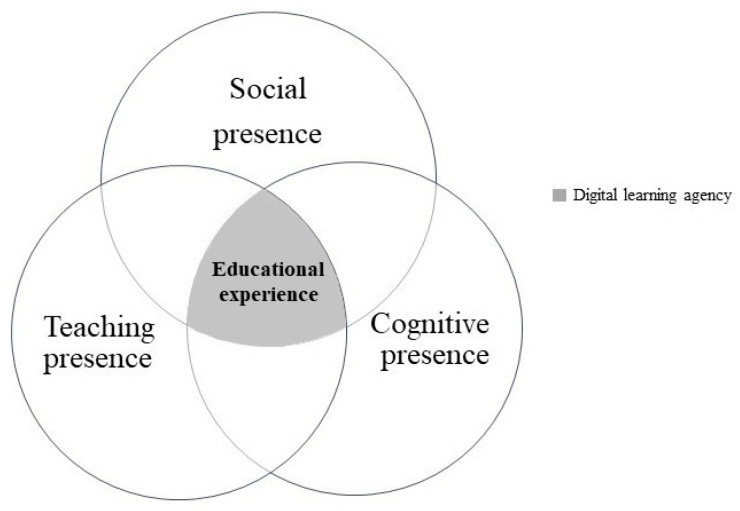
New COI version for middle school students in online environments.

**Table 1 behavsci-12-00466-t001:** The design of the online programming course.

Online Programming Course		
Platform	TalentLMS (Learning Platform)	Thunkable (Programming Platform)
Objective	students used this platform to learn and communicate	students used this platform to build their own game
Description	TalentLMS offers: guidelines and support gamification technique: points, badges and rewards digital materials: video, texts, flowchart, stages map, activities synchronous communication: virtual classroom (live sessions) with chat area and microphone asynchronous communication: online discussions learner assessment: self-assessment, quizzes	Thunkable offers: creating new project adding components (buttons, images, text…) adding voices setting layout downloading App sharing App publishing

**Table 2 behavsci-12-00466-t002:** Examples of quotes/data reforming the sub-themes and the main themes.

Themes	Sub-Themes	Quotes/Data
Rationale for Online Learning (social presence & cognitive presence)	flexibility	It is easier [the course], I can access it any time (student’s interview). An advantage of the online course for the students is the availability of accessing the materials on their convenience (teacher’s interview).
	problem-solving skills	I will benefit from learning programming. When I face problems, I will be able to solve them. Also having a certificate will make me better than others (focus group).
	time investment	It is enjoyable, and it is better than wasting my time in watching episodes (focus group).
	passion to learn programming	Get benefits and have a new experience and learn about programming (student’s interview).
	planning future careers	Probably I will pursue game design in the future (student’s interview).
Teacher’s role (Teaching presence)	orientation	Having a live orientation meeting in the beginning of the course would address the main challenges facing the students (teacher’s diary).
	timely support	Students expect timely responses to their inquiries. I think timely responses increase students’ enthusiasm for learning and their progress (teacher’s interview).
	encouragement	I noticed that using encouraging words has assisted to increase students’ motivation to keep on progress, earn badges, and compete with their peers (teacher’s interview).
Course design (Social presence, cognitive presence & teaching presence)	virtual meeting	The live sessions helped me understand the difficulties that I met in accessing the link to Thunkable (student’s interview).
	guidelines and support	This course offers guidelines and support from the teacher as well as enabling me to practice (student’s interview).
	gamification	Online learning has great advantages. It is enjoyable and interesting when it includes competitions (focus group).
	content	I like to watch the videos to practice without missing any step (student’s interview).
	game development	The course enabled me to learn enjoyable things I have learned how to develop a game I learn the fundamental issues in programming with fun as I made a game and played with it (student’s interview).
Digital learning agency (Digital confidence, digital competence & digital accountability)	technological issues	No reports of any login difficulties or problems with their TalentLMS accounts or the Thunkable platform (content analysis). Access was via laptop and mobile devices (focus group).
	student-led instruction	Self-learning skills (system’s student progress report)
	live collaboration preference	Students supported each other (collaborative online learning) during the live sessions (teacher’ diary). This (experience) simplified many steps as we helped each other when working at the same time (focus group).
	responsibility	Concern about future careers and maintaining family privacy (focus group).

## Data Availability

The data presented in this study are available on request from the corresponding author. The data are not publicly available due to privacy or ethical restrictions.

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
