# Peer review of "Experiences of Middle School Programming in an Online Learning Environment"

_behavsci, 2022, doi:10.3390/bs12110466_

Round 1

Reviewer 1 Report

This research evaluates learners’ experiences from studying computer programming in an online course using qualitative methods such as interviews, a focus group, a teacher’s diary, and a content analysis of student activities recorded in an LMS. The study also explores the influencing factors for engaging learners in programming.

In my opinion, the aim of the study is clearly communicated, the methodology followed has no flaws (the research design, questions, hypotheses, and methods are clearly stated), and the results are clearly presented. The paper is also well-written.

I only have a few recommendations to make in order to improve the paper and more specifically I think that the following points should be clarified:

On page 4, it is mentioned that “Thunkable was recommended by Common Sense”. Brief information is needed regarding Common Sense (is it an organization?).

It is mentioned that the Thunkable platform is used as an environment for learning programming and TalentLMS as a platform for gamifying the process. How do these platforms interoperate? How are the achievements in Thunkable rewarded by points or badges in TalentLMS? Is this an automated process or does it take place manually? Is Thunkable somehow integrated into TalentLMS?

Reviewer 2 Report

The paper deals with very important topic of programing learning for young students. Research is significant also because it includes girls from Saudi Arabia and emphasize importance of education of adolescent girls in that country and cultural setting.   

Structure of the paper is adequate and aim of the research is clear. Theoretical introduction and literature review is relevant and deals with contemporary trends in teaching and learning theories. Authors especially empresses learning trends in modern society like principle of gamification.  

Also, there is a room for improvement. In methodological section research question should be expanded more to include what are key limiting factors for effective learning of programing and what are possible changes on educational system level that can mitigate those problems.

In order for paper to be improved more it is advisable that phases of the research process should be described in more detail in methodological section. That includes detailed description of research process regarding individual interviews, focus group, teacher’s diary, and content analysis.

Main limitation of the research that this paper deals with very specific learning program, and because of that we should be very carefully regarding possible generalization on other similar examples in Saudi Arabia and rest of the world. 

However, this research can have very important local significance and can lead to the changes that may improve educational effectiveness, especially for girls in Saudi Arabia.  

Reviewer 3 Report

The subject of the study is an important subject and it is believed that it will contribute to the field. It is suggested that researchers explain in detail the contribution of the article to the field and to whom it will benefit.

The article needs to be systematically rearranged. For example, the subheadings under the I. Introduction section continue as 3.1. It is recommended to check systematically.

In the introduction section, the reason for the study, and the problem situation should be clearly stated.

I believe that the method part of the study is weak. It is suggested that the titles of the research model, participants, data collection tools, application and procedure of the study, data analysis techniques, and ethical permissions should be explained in detail in the method section of the article.

Round 2

Reviewer 3 Report

Reviewer suggestions were taken into consideration. Thank you

Author Response

Many thanks for your considerations